# Challenges in Combining EMG, Joint Moments, and GRF from Marker-Less Video-Based Motion Capture Systems

**DOI:** 10.3390/bioengineering12050461

**Published:** 2025-04-27

**Authors:** H. M. Rehan Afzal, Borhen Louhichi, Nashmi H. Alrasheedi

**Affiliations:** 1Key Laboratory for Space Bioscience and Biotechnology, Engineering Research Center of Chinese Ministry of Education for Biological Diagnosis, Treatment and Protection Technology and Equipment, School of Life Sciences, Northwestern Polytechnical University, Xi’an 710072, China; 2Deanship of Scientific Research, Imam Mohammad Ibn Saud Islamic University (IMSIU), Riyadh 11432, Saudi Arabia; 3Department of Mechanical Engineering, College of Engineering, Imam Mohammad Ibn Saud Islamic University (IMSIU), Riyadh 11432, Saudi Arabia

**Keywords:** marker-less motion capture, ground reaction forces, biomechanical analysis, motion capture technology, multimodal data fusion

## Abstract

The evolution of motion capture technology from marker-based to marker-less systems is a promising field, emphasizing the critical role of combining electromyography (EMG), joint moments, and ground reaction forces (GRF) in advancing biomechanical analysis. This review examines the integration of EMG, joint moments, and GRF in marker-less video-based motion capture systems, focusing on current approaches, challenges, and future research directions. This paper recognizes the significant challenges of integrating the aforementioned modalities, which include problems of acquiring and synchronizing data and the issue of validating results. Particular challenges in accuracy, reliability, calibration, and environmental influences are also pointed out, together with the issue of the standard protocols of multimodal data fusion. Using a comparative analysis of significant case studies, the review examines existing methodologies’ successes and weaknesses and established best practices. New emerging themes of machine learning techniques, real-time analysis, and advancements in sensing technologies are also addressed to improve data fusion. By highlighting both the limitations and potential advancements, this review provides essential insights and recommendations for future research to optimize marker-less motion capture systems for comprehensive biomechanical assessments.

## 1. Introduction

The development of marker-less technologies revolutionized the science of biomechanics considerably. Marker systems were traditionally applied to monitor movement by utilizing body-borne physical markers. Although very effective, these systems were constrained by the need to have exact marker placement and a well-controlled environment, so their potential applications were very specific. Marker-less systems based on computer vision and machine learning have made movement analysis easier and more adaptable. These systems, such as the Microsoft Kinect, permit real-time tracking of joints and provide both the depth and the RGB information with a less-invasive approach to analysis compared with marker systems [1]. The change to marker-free systems made them applicable to a broad range of applications, such as rehabilitation, sports science, and ergonomics, with the potential to carry out dynamic and natural movement analyses.

Integrating electromyography (EMG), the joint moments, and ground reaction forces (GRF) into the body movement analysis offers a comprehensive understanding of body movement. The EMG registers muscle activity patterns to enlighten the involvement of the muscles in movement control [2]. The joint moments are the determinants of the information about the forces at the joints to appreciate the mechanical stress experienced by the musculoskeletal apparatus [3]. The information obtained by the ground reaction forces is central to gait and balance analysis. It informs the interaction of the body with the ground to improve the overall dynamic of movement [4]. Coupling of the modality offers a comprehensive view of the body’s movement to improve the analysis of the body’s movement with greater accuracy [5].

The rationale for including EMG, joint moments, and GRF is due to the multifaceted and complex nature of the movement of the body of humans. Traditional approaches are predominantly derived from independent measurements that may fail to value the complex interactions among the involved body systems fully. Multimodal analysis enables a deeper understanding, translating to more valid evaluations and personalized interventions. It is particularly beneficial within the clinical setting, where personalized intervention protocols can be directed by a comprehensive analysis of movement patterns of individuals recovering from injury or with chronic conditions [6]. In addition to this, the multimodal approach also has potential benefits in establishing predictive models that improve the validity and reproducibility of the analyses of the body’s mechanics.

### 1.1. Objectives of the Review

This review will examine the challenges of incorporating EMG, joint moments, and GRF into video-based motion capture systems. Despite technological advancements, challenges to the accuracy of the information, cross-modal alignment, and the need for standard protocols persist. Furthermore, although marker-less systems have made strides, comparison with the traditional method is still warranted to introduce a measure of robustness and reliability across various environments. This review will elaborate on the challenges and present directions to fill the current knowledge gaps.

### 1.2. Structure of the Review

The structure of this paper is as follows. Section 2 provides an overview of the evolution of motion capture technology, emphasizing the shift from marker-based to marker-less systems and their implications for biomechanical research. Section 3 is about the methodology of review, whereas Section 4 discusses the significance of integrating EMG, joint moments, and GRF in the analysis of human movement using vide based motion system and highlighting the advantages of a multimodal approach. Section 5 addresses the challenges and limitations of current methodologies, while Section 6 evaluates existing technologies, and Section 7 proposes future research directions with emerging trends. Finally, the conclusion summarizes key findings and emphasizes the need for innovation in video-based motion capture and biomechanical assessments.

The advancements in motion capture technology, especially the changeover to marker-less systems, have progressed the science of biomechanics. Including the EMG, the moments of the joints, and the GRF increases the knowledge of the movement of the human body, which has significant implications for clinical applications. With the recognition of the current challenges and the solution to them, this review hopes to aid the establishment of more robust and thorough methodologies of biomechanics.

## 2. Background and Literature Review

### 2.1. Marker-Less Motion Capture Systems

Marker-less motion capture systems are a major improvement compared with marker-based technologies that prevailed in previous decades within the domain of biomechanics. Marker-based systems have traditionally involved the affixation of physical markers to the subject’s body that is tracked by several cameras to reconstruct the three-dimensional movement information. Although highly effective, marker-based systems are time-consuming to operate, need meticulous calibration, and are constrained by the need for a controlled environment to avoid error due to occlusions and other interference [7]. A block diagram that summarizes the marker-less system for human body pose estimation can be seen in Figure 1. Marker-less systems, by contrast, leverage the capabilities of computer vision and machine learning algorithms to obtain movement information directly from video recordings without the need for physical markers. This change enables the movement to be captured naturally while allowing greater freedom of the subject without the need to constrain the individual with markers [8]. Marker-less video-based motion system showing the main steps to estimate different forces such as GRF, Joint and muscle forces, etc., can be seen in Figure 2.

The advantages of marker-less systems are apparent: they are easier to install, have shorter setup times, and can be applied to various environments, from clinics to sports stadiums, making them more usable [9]. New sensor technologies, such as the depth camera and the inertial measurement units (IMUs), have also made marker-less motion capture more precise and reliable. Marker-less systems are still facing a number of challenges. Data quality can deteriorate with complex movement or with occlusions, notably where body parts overlap with other body parts, while sudden movement or sudden change in direction can also add inaccuracies to the tracking of movement [10]. All of this must be overcome to fully leverage the potential of marker-less systems in motion capture across applications.

### 2.2. Electromyography (EMG)

EMG is a fundamental technique used to measure the electrical activity of skeletal muscles, providing essential insights into muscle activation patterns during physical activities. These data are invaluable in clinical, sports science, and ergonomics applications, as they enable the assessment of muscle function, fatigue, and coordination. In rehabilitation, EMG helps monitor recovery, while in sports, it is used to optimize performance by analyzing muscle recruitment strategies. However, the acquisition of reliable EMG data presents several challenges. Signal interference from electrical noise, movement artifacts, and cross-talk between adjacent muscles can compromise data quality [11]. Moreover, improper electrode placement can result in significant errors, further complicating data interpretation.

EMG data analysis also requires complicated signal processing to discriminate the signal of interest from the noise to gain meaningful information. This can also limit proper usage within environments that lack the necessary expertise. Despite these limitations, EMG is a very significant way of analyzing the dynamic behavior of the muscle, although coupling with other biomechanics, such as the joint moments and GRF, is needed to present a comprehensive analysis of human movement.

### 2.3. Joint Moments

Joint moments are critical in understanding the mechanical forces acting on the body during movement. These moments, generated by muscle forces, body weight, and external loads, determine the rotational forces at the joints, which are crucial for assessing movement efficiency and stability. In biomechanics, joint moments are often calculated using inverse dynamics, which combines motion capture data with force measurements to estimate the forces and moments at the joints during movement.

Several methods of estimating joint moments are present, with their strengths and weaknesses. Conventional techniques are marker-based systems of motion capture and force plates that measure the exact value of GRF and joint moments. With the advent of marker-less systems, the potential of using computer vision and machine learning algorithms to estimate the moments of joints with the aid of video information is being researched. These approaches are more convenient and adaptable but are challenged by their accuracy and the requirement of being compared with traditional methodologies. The coupling of the information of EMG with the analysis of moments of joints can lead to a greater knowledge of the effect of the muscles’ activity on the joints’ mechanics with a deeper understanding of the movement of humans.

### 2.4. Ground Reaction Forces (GRF)

Ground reaction forces are necessary to appreciate the interaction between the ground and the body while in movement. The forces are also necessary to examine gait analysis, stability, and overall musculoskeletal loading while engaging in movements such as walking, running, and jumping. GRFs are normally collected with the aid of a force plate that measures forces applied to the ground while the body is in movement. With the aid of movement tracking information, the measurements provide significant information regarding the body’s mechanical requirements while in movement.

While force plates are the current measurement standard GRFs, valid acquisition of the GRFs with marker-less systems is still a problem. Enhanced algorithms and advancements in sensor fusion methodologies have made some marker-less systems capable of approximating the GRFs. However, the quality of the approximated measures can also be influenced by the sensors’ location, environmental conditions, and the movement being monitored. With the advancements of marker-less technologies continuing to improve, the solution to the aforementioned problems will be helpful to their successful deployment into research and clinical applications to permit the delivery of more valid and comprehensive biomechanics evaluations.

## 3. Methodology of the Review

### 3.1. Literature Search Strategy

To identify applicable studies to this review, a comprehensive literature search was undertaken with the aid of various major academic databases, namely PubMed, IEEE Xplore, and Web of Science. The databases were chosen because of their extensive coverage of research work within the areas of both biomedicine and engineering including studies of various disciplines involved with the areas of biomechanics, motion capture, EMG, joint moments, and GRF. All databases were searched to encompass both the clinical and the engineering aspects to cater to the multi-interdisciplinary aspect of marker-less motion capture systems.

The search was made with a combination of terms and operators to obtain a comprehensive retrieval of the research on the incorporation of EMG, moments at joints, GRF, and marker-less movement tracking systems. The following key terms were used:“Marker-less motion capture” AND “biomechanics”“EMG integration” AND “Marker-less motion capture“Joint moments estimation” AND “Marker-less video”“Ground reaction forces” AND “Marker-less motion capture“Deep learning” AND “biomechanics” AND “motion capture“EMG data fusion” AND “Marker-less systems

The searches were conducted on articles that were published from 2019 to 2025, with the results being confined to English-language articles.

### 3.2. Inclusion and Exclusion Criteria

The inclusion criteria were established to screen out irrelevant research and permit the inclusion of only high-quality research that was directly applicable to this review.
**Inclusion Criteria****Exclusion Criteria**Study Types: Peer-reviewed original research, review articles, and conference proceedings involving the integration of EMG, joint moments, and GRF with marker-less motion capture systems.Study Types: Articles not related to biomechanics or motion capture, case reports, opinion pieces, and studies on non-human subjects (e.g., animals, objects).Publication Dates: Studies published between 2019 and December 2025 reflect recent advancements in the field.Publication Dates: Studies published before 2018 because of outdated methodologies and technologies.Quality Standards: Studies with validated motion capture protocols, experimental controls, and advanced analysis (e.g., deep learning, biomechanical modeling). Quantitative or qualitative assessments are included.Methodological Quality: Studies lacking clear methodology, proper sample size, peer review, or evidence of integrating EMG, joint moments, and GRF with marker-less systems.

### 3.3. Limitations of the Review Process

While every possible effort was made to provide the comprehensiveness and validity of this review, several limitations must be pointed out.

Limited Generalizability of the Findings: This review covered experiments with a mix of experimental designs, groups of participants, and technologies. The diversity means that the results may not necessarily apply to all conditions in all circumstances, e.g., other sports activities or diseases.

Language Bias: The present review included only English-language studies, possibly excluding research printed in other languages. Thus, potentially significant research could have been omitted.

Exclusion of Non-Experimental Studies: Although the review focused on peer-reviewed experimental studies, clinically significant information could also emanate from the grey literature, technical documents, or clinical reports. The research excluded these, potentially excluding practical applications or case evidence.

Despite these limits, the methodology covered within this review ensures a comprehensive and systematic approach to understanding the status of research, including EMG, moments of the joints, and GRFs with marker-less movement tracking systems.

## 4. Integration of EMG, Joint Moments, and GRF: Current Trends

### 4.1. Data Acquisition Techniques

The integration of GRF, joint moments, and EMG into marker-less motion capture systems has progressed considerably with the advancements of sensor technologies and video-based approaches. IMUs, pressure sensors, and surface electromyography (sEMG) are central to the acquisition of detailed neuromuscular and biomechanical information. Muscle activation patterns are commonly assessed with the aid of sEMG to gain information regarding the function of the neuromuscular apparatus with movement. High-density approaches to EMG have since been developed to improve the spatial resolution to measure the activity of the muscle with greater detail [12]. Several key sensor technologies, with their advantages, limitations, and applications, that are used in kinetics are tabulated in Table 1.

Video-based motion tracking systems with embedded machine learning algorithms are increasingly applied to marker-less body movement tracking. The systems are equipped with cameras to record body movement and measure body movement by analyzing the body’s movement without physical markers [13]. With the fusion of computer vision with EMG information, the movement of humans can be understood with greater detail. Nevertheless, challenges are present with the systems being less accurate and less reliable with the presence of dynamic conditions that have the potential to degrade the quality of the information [14]. Camera (video-based) techniques are usually simpler, less expensive methods that utilize standard 2D or RGB cameras, i.e., camcorders or smartphones, to record motion. They tend to depend on computer vision algorithms such as 2D pose estimation or optical flow estimation and usually do not involve dedicated hardware or calibration.

Furthermore, anatomical variations and inconsistent sensor placement can introduce discrepancies, necessitating robust calibration techniques and standardization protocols to ensure data comparability across subjects and studies [15].

### 4.2. Data Fusion and Synchronization

Data fusion and time synchronization are central to the proper alignment of EMG, joint moments, and GRF data. Procedures have been put into practice to time-sync data among the various sources with time-stamping, cross-correlation, and learning algorithms that learn the time relationships between streams of information [16]. Time-stamping methodologies ensure that the acquired information from the various sensors is aligned with the time sequence of events. Cross-correlation methodologies can discover the delays among the signals to align the non-simultaneous information properly [17].

A number of techniques are being applied for registration techniques between force plates and marker-less systems. 1. Camera-to-Force Plate Co-Calibration: both force plate and marker-less system are co-calibrated with certain calibration objects (e.g., checkerboard or rig made for that purpose) that are tracked by both systems. This aligns with their coordinate frame better. 2. Visual-Aided Registration: pose estimation algorithms based on vision to detect foot location or limb-ground points of contact in video images, which are registered with the established location and known size of the force plates. Depth information or multiple views are usually required to enhance spatial accuracy. 3. Alignment based on Machine Learning: Some advanced systems make use of machine learning algorithms trained on both marker-less and marker-based data sets to learn the relationship between the two systems’ coordinates, indirectly performing registration. We provide an integrated summary that includes both the methods and relevant machine learning-based validation studies. The method used: Trains deep learning models to learn the spatial mapping between marker-based and marker-less coordinate systems. Validation: [18] reported registration errors <7 mm and joint moment RMSEs ranging from 5–12 Nm across joints, suggesting acceptable biomechanical fidelity.

Software tools and algorithms are at the forefront of enabling multimodal data fusion. Deep learning techniques have also been implemented to gain meaningful information from various streams of heterogeneous data to find complex patterns that could remain unknown if the modality was processed individually [19]. For instance, deep learning models can integrate information about the EMG signal, kinematic information of the joints, and the ground reaction forces to identify movement patterns to improve the analysis of the movement and the outcome of the tests of the rehabilitation [20]. Bayesian techniques are also probabilistic models that enhance the outcome of the fused information by considering the sensors’ intrinsic measurement uncertainties to improve the output’s robustness [21]. Several key methods used for data fusion and synchronization are detailed in Table 2.

Despite these advancements, challenges exist to have a smooth fusion of information. Sensor variability can introduce inaccuracies into the fused output with the need to develop increasingly complex algorithms that can learn to adapt to varying conditions of sensors and performance levels. Additionally, the computational intensity of real-time fusion algorithms is a limitation to applications that must provide immediate response, e.g., assistive technologies or rehabilitation [22].

High-density EMG (HD-EMG) provides a variety of benefits compared with conventional EMG systems when implemented in motion capture systems, but it also comprises some drawbacks. Some of the benefits are: 1. Increased spatial resolution, including better muscle mapping, which makes it possible to capture a more precise spatial representation of muscle activity. 2. Signal quality and noise Attenuation by signal averaging: With a dense electrode mesh, HD-EMG can utilize advanced processing techniques that average across numerous channels to eliminate the effects of random noise and increase the signal-to-noise ratio. It also comprises some drawbacks, including increased data complexity and processing requirements that make it time consuming because of high dimensional data. Another limitation is practical and technical challenges, including setup and electrode placement and costly equipment load.

### 4.3. Validation Strategies

Validating marker-less systems is necessary for their validity and comparability to established techniques, such as using force plates and wired EMG systems. It is common practice to compare the marker-less systems with the results of established systems to estimate the level of correspondence [23]. It is known that marker-less systems can provide estimates of the angle of joints and moments equal to optical systems of movement if compared with them. Nevertheless, they can have inaccuracies with dynamic conditions.

Cross-validation techniques are another robust method that uses a distinct group of trials to train the model and another to validate the model. It eliminates the possibility of overfitting and helps the model to generalize to new data. The implementation of standard protocols for the processing and acquisition of data also maximizes the comparability of the results among the studies to improve the consistency of the results.

Integrating EMG measurements with traditional measures of biomechanics is a robust paradigm to cross-validate marker-less systems. Examining the correspondence between patterns of muscle activations and moments at the joints can provide evidence of the functional meaning of measurements obtained with marker-less systems. The multi-prong approach to validating marker-less systems not only maximizes their validity but also maximizes their usability within research and clinical settings.

The integration of EMG, joint moments, and GRF by means of state-of-the-art acquisition techniques, optimal data fusion, synchronization strategies, and rigorous validation protocols represents a major advance within the domain of biomechanics. With the continuing advancements of technology, the potential of marker-less systems to deliver real-time, valid information about movement will broaden to support innovative applications within the realms of both clinical rehabilitation and sports performance. Future advancements in the algorithms of data processing, sensor technologies, and protocols of validation will remain key to the improvement of the precision and trustworthiness of multimodal tests of biomechanics.

## 5. Key Challenges in Combining Modalities

### 5.1. Technical and Computational Challenges

A key technical challenge in combining modalities such as EMG, joint moments, and GRF lies in synchronizing data captured at different sampling rates. Each modality typically operates at distinct frequencies, creating temporal misalignments that complicate the integration process. For example, EMG signals are often sampled at high rates (e.g., 1000 Hz), while motion capture systems may operate at lower frequencies (e.g., 60 Hz). This discrepancy can lead to inaccuracies in data fusion and potential misinterpretations of movement dynamics. Methods such as interpolation and resampling can mitigate synchronization issues but may introduce additional uncertainties to the analysis [24].

Another challenge is the computational complexity involved in processing and fusing high-dimensional data from multiple modalities. The integration of data from EMG, joint moments, and GRF requires advanced algorithms capable of managing the intricacies of each data type while preserving data integrity. Machine learning and statistical models are frequently employed for data fusion; however, these algorithms are sensitive to noise and demand significant computational resources for training and validation. Additionally, the underlying biomechanics, which can vary considerably across different populations and activities, further complicates the generalizability of these models.

Signal noise and interference present a pervasive challenge in biomechanical data collection, especially in marker-less systems that rely on video-based motion capture. Environmental factors such as lighting, occlusions, and reflections can degrade data quality and introduce significant errors in motion analysis. Researchers use advanced signal processing techniques such as Kalman filtering and wavelet transforms to address these issues to clean data before analysis. Furthermore, optimizing the capture environment by controlling lighting and minimizing occlusions can help reduce external noise and improve data quality [25].

### 5.2. Accuracy and Reliability Concerns

Accuracy and reliability are crucial when combining modalities, especially when estimating EMG, joint moments, and GRF from marker-less systems. Measurement errors can arise from a variety of sources, including sensor calibration, environmental conditions, and the algorithms used for data processing. For instance, inaccuracies in the estimation of joint moments can substantially affect the assessment of movement dynamics, leading to erroneous conclusions about an individual’s biomechanics. Additionally, while marker-less systems offer convenience, they may not match the precision of traditional marker-based systems, which are widely considered the gold standard in motion analysis.

Calibration issues further complicate the reliability of combined modality data. Variability in sensor placement and alignment can introduce discrepancies between subjects and across different testing environments, making it difficult to compare results consistently. The lack of standardized calibration protocols in the current literature exacerbates this issue. Developing robust and easy-to-implement calibration methods will be essential for ensuring consistent data quality across diverse settings and enhancing the reliability of biomechanical assessments using marker-less systems.

### 5.3. Standardization and Validation Challenges

The absence of standardized protocols for integrating multiple data sources poses a significant barrier to the advancement of multimodal biomechanical analysis. Without established guidelines, researchers may adopt varying methodologies, making it challenging to compare findings across studies. This lack of uniformity hinders the development of a cohesive body of knowledge on human movement, as results may not be directly comparable because of methodological discrepancies. Additionally, validating marker-less systems against traditional “gold standard” methods present an ongoing challenge. While marker-less technologies offer greater accessibility and usability, their accuracy and reliability must be rigorously tested against established techniques to ensure their validity. This validation requires extensive testing across a range of activities and populations to confirm that marker-less systems can produce results comparable to those obtained from traditional methods. Establishing clear validation criteria and benchmarks will be crucial for fostering confidence in the use of marker-less systems in both research and clinical applications.

### 5.4. Environmental and Practical Constraints

Environmental factors, such as lighting conditions and occlusions, can significantly affect the accuracy of video-based motion capture systems. Poor lighting can degrade image quality, making it challenging for algorithms to detect and track movements accurately. Similarly, occlusions, when parts of the body are blocked from the camera’s view, can lead to incomplete data, resulting in inaccuracies during motion analysis. To mitigate these issues, optimal environmental setup is crucial, including controlling lighting and positioning subjects to minimize occlusion risks [26].

Moreover, the real-world applicability of marker-less motion capture systems is constrained by the dynamic nature of uncontrolled environments. While laboratory settings allow for controlled conditions that optimize data collection, real-world environments present unpredictable challenges that can affect data accuracy. Variations in terrain, unanticipated subject movements, and interactions with other individuals can complicate data analysis, requiring algorithms that can adapt to these dynamic conditions. Developing robust and adaptive algorithms will be essential for the practical deployment of marker-less systems in real-world scenarios.

### 5.5. Interdisciplinary Integration

The interdisciplinary nature of combining biomechanics, computer vision, and signal processing introduces unique challenges, particularly regarding differences in methodologies and terminologies. Researchers from diverse fields often employ distinct terminologies and approaches, which can lead to misunderstandings and inefficiencies. For example, biomechanists may focus on the physiological aspects of movement, while computer scientists prioritize algorithmic efficiency and data processing techniques. Bridging these disciplinary gaps requires collaborative efforts to establish a common language and framework for integrating knowledge across fields.

Furthermore, integrating methodologies from different disciplines is complex, as each field may follow its own validation processes and standards. For instance, validation techniques used in biomechanics may differ from those in computer vision, complicating the development of unified protocols for data collection and analysis. Encouraging interdisciplinary collaboration and fostering a shared understanding of methodologies will be essential for overcoming these barriers and advancing the integration of modalities in biomechanical research.

## 6. Case Studies and Comparison

### 6.1. Overview of Notable Studies

Several notable studies have attempted to integrate EMG, joint moments, and GRF using marker-less motion capture technologies, demonstrating the growing interest in multimodal approaches to biomechanical analysis. One notable study by Wang et al. explored using the Microsoft Kinect, a marker-less system, for upper limb rehabilitation in stroke patients. This study highlighted the feasibility of combining EMG data with motion capture to assess muscle activation patterns in conjunction with joint movements during rehabilitation exercises [27]. The results emphasized that integrating these modalities offered a more comprehensive view of the rehabilitation process, facilitating personalized interventions based on individual patient needs.

Another study by Wenqi et al. examined a novel deep-learning model to estimate the EMG envelope using IMUs with high accuracy during gait. They also developed a data augmentation-based method to improve the performance of the estimation model with small-scale datasets. They compared this method with other existing methods and suggested better results with Pearson correlation coefficient and normalized root-mean-square errors of 0.72 and 0.13, respectively. [28].

Furthermore, Whatling et al. reviewed the use of motion capture technologies in evaluating the biomechanics of osteoarthritis. This study emphasized the utility of combining GRF measurements with joint moment analysis to assess functional outcomes in patients with osteoarthritis [29]. By leveraging marker-less motion capture systems, the researchers gathered data in naturalistic environments, improving the ecological validity of their findings.

Kanko et al. tested the reliability of marker-less motion capture systems for gait analyses among clinical populations. Their 2021 study confirmed the validity of a deep learning-assisted video capture system compared with standard marker-based systems, with high agreement between joint kinematics between both systems [30]. The findings are an argument for integrating marker-less solutions in standard clinical evaluations, especially among patients with motion impairments, since these diminish preparation time and enhance patient comfort.

Another study evaluated upper-limb kinematic measurement with a marker-less motion capture system (2D cameras + Theia3D (v2021.2)) against a marker-based system. Three elite boxers underwent shadow boxing trials, which were captured by 12 optoelectronic and 10 cameras. The results indicated greater discrepancies in 3D joint center locations at the elbow (more than 3 cm) than at the shoulder and wrist (less than 2.5 cm). Joint angles had poorer agreement, particularly for internal/external rotation, but the shoulder joint was best. Segment velocities had a good-to-excellent agreement. This study implies that marker-less systems present an efficient alternative for performance analysis in dynamic environments [31].

### 6.2. Comparative Evaluation

The reviewed research showcases both the strengths and weaknesses of integrating the above modalities to conduct biomechanical analysis. Among the major strengths of marker-less systems is their capacity to allow the acquisition of data within various real-world environments. The flexibility is of significant value to applications within clinics where the traditional laboratory environment might not be feasible. Additionally, the concurrent acquisition of the GRF and the moments at the joints enables a comprehensive analysis of movement kinematics that is important to the analysis of complex interactions between the body’s components, as illustrated by the work of [32] on the analysis of ergonomics.

Despite these gains, various challenges persist. A significant limitation is the validity of information obtained from marker-less systems that are vulnerable to environmental conditions such as light levels, occlusions, and processing errors by algorithms. For example, while the Kinect system provides rich movement information, tracking of fast or complex movement is limited by its accuracy with the potential to introduce errors in estimates of the moments of GRFs. Subsequently, the lack of standard protocols to integrate information between the various data modalities also makes cross-study comparison problematic with the inability to build definitive conclusions about the effectiveness of marker-less systems to perform biomechanical analysis.

### 6.3. Lessons Learned and Best Practices

Several key learnings and best practices have emerged from research that can inform the challenges raised within the earlier sections. First, the worth of strong marker-less systems that are comprehensively validated against existing techniques cannot be overstated. Having transparent validation protocols and benchmarks is key to boosting confidence in the quality and accuracy of marker-less system data. Comparison studies of the performance of marker-less technologies across a range of settings will contribute to a greater evidence base regarding their effectiveness.

Second, including the latest signal processing algorithms can substantially improve the quality of the information captured by movement tracking systems and EMG. Maximizing signal quality by employing intricate filtering algorithms to purge the signal of noise is necessary to properly interpret the movement of joints and the patterns of muscle activations. Machine learning algorithms that aid the fusion of information can also permit the extraction of significant information from the big-dimensional information to aid a finer analysis of human movement.

Finally, fostering cross-domain interaction between the research communities of signal processing, computer vision, and biomechanics is the key to propelling the field to the next level. Having a common paradigm and jargon to bridge knowledge between the disciplines will permit the establishment of stronger methodologies to unite the different types of data. It will not merely enhance the quality of research work but also the translation of research into practice to the advantage of both the patient and the practitioner.

## 7. Emerging Trends and Future Directions

### 7.1. Advances in Machine Learning and AI

Integrating machine learning and deep learning into biomechanics research has immense potential to enhance information fusion, improve results quality, and develop predictive models. The newest advancements of deep learning, a sub-type of ML, have demonstrated the potential to operate with complex, high-dimensional information that is highly beneficial while fusing the information of GRF, EMG, and marker-less motion capture systems’ joint moments. These techniques are highly proficient at identifying complex patterns and relationships within big sets of information to develop complex models of the movement of humans and performance. For example, deep learning algorithms can be implemented to forecast the moments of joints with the support of the information on EMG signals and the information on GRFs to potentially enhance the analysis of movement dynamics and injury threats.

Machine learning also enables automation of the processing and analysis of data with minimal need for extensive hand intervention and specialized knowledge of complex datasets. It is especially useful within the medical environment, where immediate biomechanics information can guide treatment protocols to improve patient outcomes. With the advancements in machine learning techniques, their applications within the domain of biomechanics will also broaden to provide novel opportunities to tailor rehabilitation protocols to the needs of the individuals involved and enhance sports performance. It uses state-of-the-art AI technology that captures body posture and joint movements in real-time and delivers valuable biomechanics, sports science, and healthcare insights.

### 7.2. Real-Time Feedback and Analysis Systems

Advancements in real-time processing technologies will revolutionize biomechanics information applications in sports and medical environments. The ability to record and analyze information in real-time facilitates immediate feedback to enhance the efficacy of interventions. For instance, systems that combine marker-less movement tracking with real-time analysis of the EMG can provide real-time feedback about muscle activation patterns while providing therapy to permit clinicians to adapt treatment protocols on the fly.

Moreover, real-time analysis is of immense value to sports training, as it allows the coach to make immediate adjustments to athletes’ techniques and strategies based on the information provided by biomechanics. Not only is this performance-enhancing but also injury-prevention by identifying potentially hazardous movement patterns before they lead to injury [33]. With the technologies continuing to develop the way they are, their integration into everyday clinical practice and sports performance measurement will increasingly occur, with the resultant effect of the advancements being felt across both domains [34].

### 7.3. Sensor and Algorithmic Upgrades

Innovations in sensing technologies and processing algorithms are central to biomechanics science. New technologies for depth cameras and IMUs have made marker-less systems of human movement tracking much more accurate. With the support of advancements in the algorithms for processing information, it is possible to remove the noise and enhance the quality of information obtained to a greater extent [35].

Machine learning algorithms can also be applied to automatically find and compensate for the most common causes of error, such as occlusions and illumination variability, that can impact the quality of the captured movement. Using the latest statistical methodologies can also enhance the robustness of the analysis of collected data to the point that results are valid across various environments and conditions. With advancements in the quality of sensors and algorithms to process the information, the possibilities of providing increasingly accurate and valid biomechanics measurements will expand to include increased applications within the sports and medical arenas.

### 7.4. Standardization Activities

Efforts to standardize protocols to encompass the inclusion of EMG, joint moments, and the ground reaction forces into marker-less systems of motion analysis are imperative to develop the next phases of research into biomechanics. The inability to standardize methodologies has rendered the comparison of results between studies a significant issue that impedes drawing conclusions regarding the effectiveness of the methodologies [36]. The development of measures with standardizable protocols to inform data acquisition, processing of data, and analysis of results will improve the quality of research and practice by increasing the rigor and reproducibility of studies.

Organizations such as the International Society of Biomechanics and the American Society of Biomechanics are being proactive about standardization of the measurement of biomechanics, with the inclusion of a range of data modalities. In the development of all-inclusive frameworks to inform both the research and the practice communities, this will promote cross-discipline knowledge transfer and interaction. Not only will the usability and quality of the research improve with the standard protocols, but also the quality of the multimodal fusion will improve to deliver a stronger outcome to both sport performance and the practice of sport [37].

### 7.5. Interdisciplinary Teamwork

Biomechanics is a fundamentally multidisciplinary science that incorporates knowledge drawn from engineering, clinical research, sports science, and computer science. The cross-fertilization of knowledge among the various disciplines creates numerous opportunities to improve the methodologies applied to the fusion of EMG, joint moments, and GRFs. An alliance between the work of the biomechanist and the computer scientist can produce increasingly complex algorithms to carry out data fusion with greater fidelity and reliability [38].

Moreover, interdisciplinary interaction can bridge the gap between practice and research, allowing advancements in technology and methodology to be properly applied to improve patient outcomes. With the advancements in the science of movement analysis, a culture of interaction and knowledge exchange will be necessary to manage the complex challenges of movement analysis [39]. If they collaborate with each other, the researchers and the practice communities can innovate to enhance the usage of multimodal biomechanical analysis clinically and sports-wise.

## 8. Conclusions

This review has provided a comprehensive analysis of the inclusion of EMG, joint moments, and GRF within marker-less motion capture systems. The overall purpose of this paper was to discuss the methodologies being applied to include the aforementioned modalities. The integration of EMG, joint moments, and GRF offers significant benefits for understanding human biomechanics, with applications in both clinical rehabilitation and athletic performance analysis. However, the combination of these modalities is not without its challenges. These include technical issues related to data synchronization, computational complexity, signal noise, and reliability concerns. This review also pointed to the need for multi-field collaborative research among signal processing, biomechanics, and computer vision domains. Better algorithms with increased sophistication, advancements in sensing technology, and the deployment of machine learning principles to information fusion are needed to overcome the systems’ existing limits and enhance their accuracy and real-time usability. Additionally, standardization of protocols to conduct multimodal fusion could significantly improve the comparability of results among various applications and studies. In conclusion, significant work has been conducted to add EMG, joints’ moments, and GRFs to marker-less systems of motion capture, although work is still needed to enhance the existing techniques. Future research on machine learning algorithms, sensors, and algorithms will have significant potential to overcome the current challenges and expand the combined systems’ applications to sports and clinics.

## Figures and Tables

**Figure 1 bioengineering-12-00461-f001:**
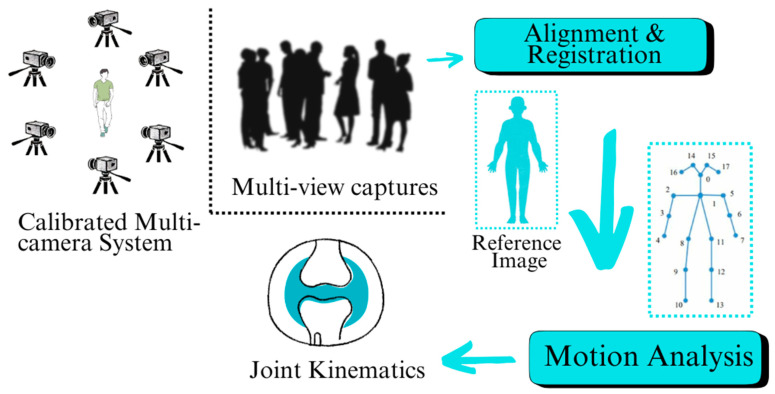
A block diagram summarizes the marker-less system for human body pose estimation.

**Figure 2 bioengineering-12-00461-f002:**
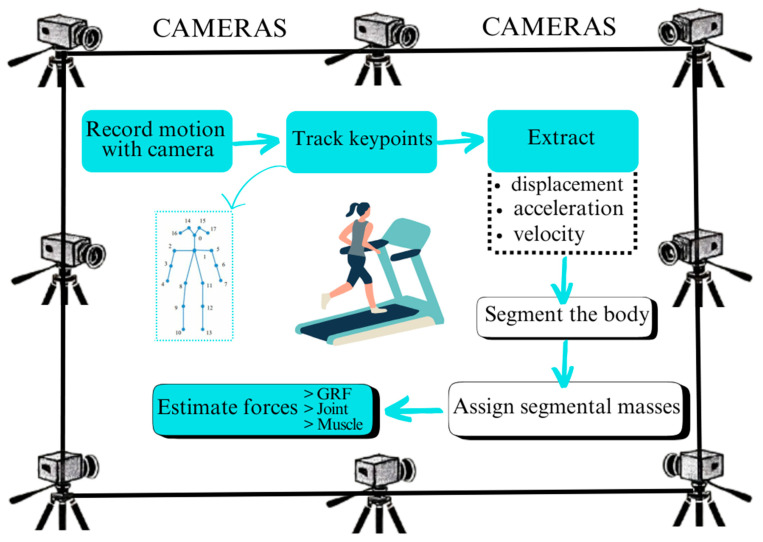
Marker-less video-based motion system showing the main steps to estimate different forces such as GRF, joint and muscle forces, etc.

**Table 1 bioengineering-12-00461-t001:** Overview of Key Sensor Technologies with their advantages and limitations used in Kinetics.

Sensor Technology	Advantages	Limitations
**Marker-less Optical Motion Capture (AI-Based Vision Systems)**	No need for markers, and it allows natural movement, growing accuracy with AI advancements	Lower accuracy than marker-based systems, affected by lighting and occlusions
**Inertial Measurement Units (IMUs)**	Portable, cost-effective, works in any environment, no occlusion issues	Drift over time requires calibration with lower accuracy for high-speed movements
**Electromagnetic (EM) Sensors**	High accuracy in small spaces, works in occluded environments	Sensitive to metal interference, limited range, expensive
**Force Plates**	Highly accurate ground reaction force (GRF) measurement, widely used in gait analysis	Expensive, limited to lab-based setups, restricted to specific force-measuring areas
**Pressure Sensors (In-Shoe, Mats)**	Portable, real-world application used in gait and plantar pressure analysis	Limited to contact forces, lower resolution compared with force plates
**Ultrasound-Based Motion Tracking**	High-resolution soft tissue imaging, useful for muscle dynamics	Limited tracking volume requires trained operators, real-time tracking challenges
**Lidar and Time-of-Flight Sensors**	Marker-less tracking, high-resolution depth sensing, and works in various lighting conditions	Expensive, limited in high-speed motion, sensitive to reflectivity
**Cameras (Video-based)**	High spatial resolution Can capture a wide range of body movements. Non-intrusive and provides a detailed 3D model.	Requires controlled lighting and environment Occlusion issues with body parts obstructing cameras.

**Table 2 bioengineering-12-00461-t002:** Key Methods Used for Data Fusion and Synchronization.

Method	Data Sources Integrated	Advantages	Challenges	Typical Application
**Kalman Filtering**	sEMG, IMUs, Force Plates, Video-based Data	- Real-time data processing - Can handle noisy data - Reduces errors by predicting future states.	- Computationally expensive - Requires accurate initial models.	- Real-time gait analysis - Sports performance optimization.
**Cross-Correlation**	IMUs, Force Plates, Video-based Data	- Simple to implement - Effective for matching time-series data - Can align data from sensors with different sampling rates.	- May be less effective in dynamic conditions - Sensitive to noise.	- Synchronizing video data with force data - Motion analysis.
**Canonical Correlation Analysis (CCA)**	EMG, Joint Moments, GRF	- Good for finding relationships between multimodal signals - Useful in identifying underlying patterns.	- Requires large datasets for accurate analysis - Can be sensitive to outliers.	- Multimodal signal integration - Clinical gait analysis.
**Independent Component Analysis (ICA)**	EMG, IMUs, Force Plates	- Separates mixed signals into statistically independent components - Handles noisy signals well.	- Complex to interpret - Sensitive to the number of sources assumed.	- Sensor signal extraction - De-noising in motion capture.
**Machine Learning Models (e.g., Neural Networks)**	sEMG, IMUs, GRF, Video-based Data	- Capable of handling large datasets - Can learn complex patterns in multimodal data - Improves with more data.	- Requires a significant amount of training data - Computationally intensive.	- Predictive modeling in biomechanics - Personalized rehabilitation models.
**Phase-locked Loop (PLL) Method**	sEMG, IMUs, Video-based Data	- Effective for aligning time-series data with distinct features (e.g., gait cycle) - Robust in real-time applications.	- Limited by the quality of initial synchronization. - Requires clear cyclical signals.	- Gait analysis - Real-time motion feedback systems.

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
