# Peer review of "Challenges in Combining EMG, Joint Moments, and GRF from Marker-Less Video-Based Motion Capture Systems"

_bioengineering, 2025, doi:10.3390/bioengineering12050461_

Round 1
Reviewer 1 Report
Comments and Suggestions for Authors
The reviewer understood the present paper provided the literature review on the advanced technologies for motion capturing, especially integration of EMG, Joint Moments, and GRF. The topic of this paper is worthy for readers of this journal. Methodology of literature survey were clearly mentioned, and survey results were well organized to clarify the recent challenges. To improve the quality of the paper, I have some comments as follows:
(1) Table 1
What is different between Marker-less optical motion capture and Camera (video-based)?
Please add further description of the method of Camera (video-based).
(2) Tabel 1
The word "kinematic" indicates mechanics concerned with motion without reference to force or mass. The table includes force plates and pressure sensors, therefore the word "kinetics" is suit for the title of this table.
(3) Registration issues between Motion-capture systems and GRF
The reviewer understood that the issues on temporal synchronization among signals (Mocap, EMG, Joint moments and GRF) were well mentioned in this survey. Whereas, spatial registration between systems (Mocap and GRF) are important as well. Based on traditional surface marker mocap systems, spatial registration between the mocap system and force platforms was done at the calibration of the mocap system. Is there any advanced registration methods between marker-less mocap and force platforms? Please mention this issues based on your literature survey.
(4) Validation study for the registration quality
Validation study for the spatial registration quality should be mentioned as well. Considering the accuracy of joint moments, the registration error between mocap and GRFs. When you add the advanced methods of spatial registration (above mentioned), please add validation study for the registration quality.
Reviewer 2 Report
Comments and Suggestions for Authors
This review is a comprehensive and valuable study with the potential to fill an important gap in the literature. However, the high content density and linguistic expression issues in certain sections may make it difficult for readers to maintain focus. In particular, the case studies, future directions, and proposed solutions should be elaborated in more depth, the language should be simplified, and repetitions should be minimized.
The abstract successfully highlights the importance of the topic, but it is overly long and contains some redundant statements. Some sentences could be made clearer and more focused.
The introduction is well-structured; however, some sentences include excessive and indirect phrasing in English. For example:
“Marker-less systems that are established on the foundations of the principles of computer vision…”
There are also some repetitions among the subheadings. For instance, the advantages and challenges of EMG are similarly discussed in both Sections 2.2 and 4.1.
The methodology section should be more concise. Information such as the number of included studies, the total number of articles analyzed, and a summary table of the data could be added to support this section.
The case studies section is underdeveloped. Only a few studies are referenced. This part should be enriched with more examples and detailed analyses.
The conclusion section summarizes the general scope of the study, but again, it is quite lengthy and contains repetitive sentences.
Reviewer 3 Report
Comments and Suggestions for Authors
Recommendation: Major revision
The review presents a comprehensive overview of the integration of EMG, joint moments, and GRF into marker-less video-based motion capture systems. However, significant revisions are necessary to improve the clarity and depth of the discussion, particularly concerning the challenges associated with data synchronization and algorithm complexity. The review should further elaborate on the specific limitations of current sensor technologies and provide more detailed comparisons between marker-based and marker-less systems, especially regarding accuracy and reliability in dynamic conditions. Additionally, the paper would benefit from a more robust analysis of the emerging machine learning techniques and their practical applications in overcoming the highlighted challenges.
- How does the review ensure that the integration of EMG, joint moments, and GRF into marker-less motion capture systems has been evaluated thoroughly across various biomechanical applications?
- Could the paper provide more detail on how the challenges related to data synchronization between EMG, joint moments, and GRF were addressed in the reviewed studies?
- What are the specific limitations of using marker-less systems in dynamic environments, and how might they be mitigated in future research?
- Are there particular case studies that were found to be more successful than others in combining EMG, joint moments, and GRF data?
- What is the impact of environmental factors such as lighting and occlusion on the accuracy of marker-less motion capture systems, and how do current studies address this challenge?
- elaborate on the advantages and limitations of high-density EMG in motion capture systems compared to traditional EMG systems?
- How could advancements in machine learning and AI further enhance the integration of EMG, joint moments, and GRF in biomechanical research?
- What advancements in sensor technology are most promising for improving the accuracy of GRF measurements in marker-less systems?
Round 2
Reviewer 2 Report
Comments and Suggestions for Authors
The authors made the suggested corrections, thank you. But they did not use the journal's template in the revision? The article should be corrected according to the template.
Reviewer 3 Report
Comments and Suggestions for Authors
Accept in present form